# Solving Inverse Problems with Model Mismatch using Untrained Neural Networks within Model-based Architectures

**Peimeng Guan**                                                                  *pguan6@gatech.edu*
*Department of Electrical and Computer Engineering, Georgia Institute of Technology*

**Naveed Iqbal**                                                            *naveediqbal@kfupm.edu.sa*
*Department of Electrical Engineering and Interdisciplinary Research Center for Communication Systems and Sensing (IRC-CSS), King Fahd University of Petroleum and Minerals*

**Mark A. Davenport**                                                              *mdav@gatech.edu*
*Department of Electrical and Computer Engineering, Georgia Institute of Technology*

**Mudassir Masood**                                                         *mudassir@kfupm.edu.sa*
*Department of Electrical Engineering and Interdisciplinary Research Center for Communication Systems and Sensing (IRC-CSS), King Fahd University of Petroleum and Minerals*

**Reviewed on OpenReview:** *https://openreview.net/forum?id=XHEhjDxPDl*

## Abstract

Model-based deep learning methods such as *loop unrolling* (LU) and *deep equilibrium model* (DEQ) extensions offer outstanding performance in solving inverse problems (IP). These methods unroll the optimization iterations into a sequence of neural networks that in effect learn a regularization function from data. While these architectures are currently state-of-the-art in numerous applications, their success heavily relies on the accuracy of the forward model. This assumption can be limiting in many physical applications due to model simplifications or uncertainties in the apparatus. To address forward model mismatch, we introduce an untrained forward model residual block within the model-based architecture to match the data consistency in the measurement domain for each instance. We propose two variants in well-known model-based architectures (LU and DEQ) and prove convergence under mild conditions. Our approach offers a unified solution that is less parameter-sensitive, requires no additional data, and enables simultaneous fitting of the forward model and reconstruction in a single pass, benefiting both linear and nonlinear inverse problems. The experiments show significant quality improvement in removing artifacts and preserving details across three distinct applications, encompassing both linear and nonlinear inverse problems. Moreover, we highlight reconstruction effectiveness in intermediate steps and showcase robustness to random initialization of the residual block and a higher number of iterations during evaluation. Code is available at `https://github.com/InvProbs/A-adaptive-model-based-methods`.

## 1 Introduction

Consider an inverse problem of the following form:

$$\boldsymbol{y} = \mathcal{A}(\boldsymbol{x}) + \boldsymbol{\epsilon}. \tag{1}$$

The goal is to reconstruct the latent signal $\boldsymbol{x}$ from the measurements $\boldsymbol{y}$ in the presence of noise $\boldsymbol{\epsilon}$, where typically the forward model $\mathcal{A}$ is assumed to be known. Inverse problems are generally challenging because they are often ill-posed, *i.e.,* the solution is not unique, or the reconstruction is highly sensitive to noise and/or model mismatch. The traditional approach to recovering $\boldsymbol{x}$ from the measurements $\boldsymbol{y}$ is by solving a regularized optimization problem of the form:

$$\min_{\boldsymbol{x}} \frac{1}{2} \|\boldsymbol{y} - \mathcal{A}(\boldsymbol{x})\|_2^2 + \gamma r(\boldsymbol{x}), \tag{2}$$

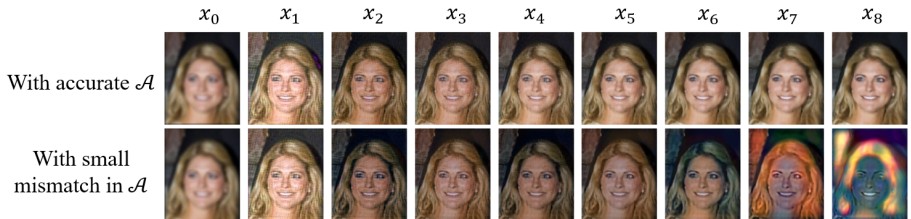

Figure 1: A proximal LU network is trained for a deblurring task using a *single* forward model. The top row shows the intermediate reconstructions over 8 iterations using the true model, while the bottom row shows the evaluation results when a small perturbation is added to the forward model (the Peak Signal-to-Noise Ratio of the true kernel to the noisy kernel is 40.9 dB). This quality degradation is due to the accumulation of errors in the forward model.

where $\gamma \geq 0$ is an appropriately-chosen parameter. The regularizer $r$ is usually predetermined based on some known or desired structure, e.g., $\ell_1$-, $\ell_2$-, or total variation (TV) norm to promote sparsity, smoothness, or edges in image reconstructions, respectively. Solving (2) requires careful consideration of the underlying physics or the forward model $\mathcal{A}$ to obtain a stable and accurate reconstruction. However, knowing $\mathcal{A}$ can be challenging in practice. The reasons for this include inaccurate measurements and challenging calibration, highly nonlinear and/or computationally expensive models replaced by simplified versions, as well as access to only approximations of certain features. Usually, some knowledge of the true model, designated $\mathcal{A}_0$ in this work, is available. This occurs in many applications, including blind deconvolution/deblurring problems, recovering seismic layer models using the simplified acoustic wave equation as the forward model (Mousa & Al-Shuhail, 2011), determining fault locations in media with unknown structures (Mahmoud & Khalid, 2013), computational tomography (CT) when the dynamic behavior of a human object is observed during the measurement period Blanke et al. (2020), sparse signal recovery when the sensing matrix is not known perfectly, etc.

When the forward model is precisely known, the inverse problem in (2) can be solved via classical optimization methods, where $r$ is predefined. Machine-learning approaches, as summarized in Arridge et al. (2019); Iqbal et al. (2023); Ongie et al. (2020), have demonstrated superior reconstruction performance by effectively learning a regularizer from data. For example, the Plug-and-Play method (Venkatakrishnan et al., 2013) trains a general denoiser independently of the forward model and iteratively minimizes (2) with the learned denoiser as the regularization updates. Another class of approaches, the *loop unrolling* (LU) method (Gregor & LeCun, 2010; Hershey et al., 2014; Adler & Öktem, 2018) and its extension using deep equilibrium models (Gilton et al., 2021a), build on the observation that many iterative algorithms for solving (2) can be re-expressed as a sequence of neural networks (Gregor & LeCun, 2010), which can then be trained to remove artifacts and noise patterns associated with a known $\mathcal{A}$, resulting in higher quality reconstructions.

While these model-based machine learning solvers have demonstrated impressive performance, they encounter challenges while dealing with inaccurate forward models. Plug-and-Play, LU, and its deep equilibrium extensions all entail a gradient update of the data-fidelity term, which can be in error when the forward model is inaccurate. This causes the learned denoiser or regularizer to be ineffective. In particular, while LU often exhibits state-of-the-art performance, has faster runtime, and has improved stability in training (Guan et al., 2022) compared to other approaches, it requires precise knowledge of the forward model. Fig. 1 demonstrates the sensitivity of LU to mismatch in $\mathcal{A}$.

Some works passively address errors in the $\boldsymbol{y}$-space by enhancing the robustness of the neural network. For example, Krainovic et al. (2023) suggests that introducing jittering in the $\boldsymbol{y}$-space can achieve robustness in worst-case $\ell_2$ perturbations. In Hu et al. (2023), a deep equilibrium solver is trained using various incorrect $\mathcal{A}$'s, demonstrating greater robustness to variations in $\mathcal{A}$ compared to the Plug-and-Play method. However, it is noteworthy that such robust networks often experience a tradeoff between accuracy and stability, as discussed in Krainovic et al. (2023); Gottschling et al. (2023). In contrast, our proposed approach involves an active strategy for mitigating model mismatch. The experimental results in Section 6 illustrate the effectiveness of our method compared to the same LU network that merely learns a robust mapping passively.

Various active forward model matching algorithms are tailored to different tasks. For linear IPs, model mismatches can be more easily resolved by alternatively updating the model parameters and the underlying signal. For instance, Fergus et al. (2006); Cai et al. (2009); Levin et al. (2009); Cho & Lee (2009) reconstructed the latent images with updates in the forward model solved directly from least squares. However, these methods are often constrained to linear scenarios and rely on task-specific recovery techniques, which are typically highly sensitive to hyperparameters Cho & Lee (2009). Later works Nan & Ji (2020); Gilton et al. (2021b) learn a regularizer from data, resulting in improved performance on linear inverse problems. The structure of the initial forward model in a nonlinear inverse problem often differs from the true model. For instance, linear models may be used to estimate nonlinear functions. This motivates the use of neural networks to learn the true forward model (Candiani et al., 2021; Koponen et al., 2021; Lunz et al., 2021). However, traditional neural network training approaches require separate stages for learning the forward model and signal reconstruction. This requires a substantial amount of training data to ensure accurate mapping (Arjas et al., 2023) from any $x$ (which represents ground truth and intermediate reconstructions) to its corresponding $y$ in iterative reconstructions. With limited training data, this method falls out of the discussion.

In contrast, we introduce an untrained residual network capable of handling forward model mismatch for general IPs. The proposed method is a unified framework that applies to both linear and nonlinear IPs, offering enhanced stability during training, requiring no additional data, and allowing fitting the forward model and reconstruction simultaneously in a single pass. We further list our contributions as follows:

- **General model-based architecture in handling model-mismatch:** We present a general model-based architecture for solving IPs when only an approximation of the forward model is available. Unlike classical model-based architectures which require precise knowledge of $\mathcal{A}$, we propose two variants of the model-based algorithms that can iteratively update the forward model along with the reconstruction.
- **Introducing untrained neural network for model-mismatch:** We introduce a novel approach by integrating an untrained neural network into a model-based architecture to handle model-mismatch in both linear and nonlinear inverse problems. By updating its parameters alongside the reconstruction process, we circumvent the need for additional data and pre-training the forward model in solving nonlinear inverse problems. Indeed, we show that random initialization of this network during evaluation yields the same level of reconstruction.
- **Proof of convergence and empirical validation:** We establish the convergence of the proposed algorithm under mild conditions and empirically verify its convergence using a Deep Equilibrium Model (DEQ) structure.
- **Improved performance in linear and nonlinear IP tasks:** In contrast to the passive robust training approaches within general IP solvers using model-based architectures, our methods demonstrate substantial improvement in image blind deblurring (linear), seismic blind deconvolution (linear), and landscape defogging (nonlinear) tasks. It's worth noting that while our application showcases improvements in these specific areas, its scope extends beyond to fields like medical imaging.
- **Effectiveness in iterative reconstructions and robustness:** We show that proposed algorithms lead to more effective intermediate reconstructions. They also exhibit robustness to different random initializations of the residual block and are more stable in scenarios involving a higher number of iterations during evaluation.

## 2 Related Works

### 2.1 Review of Model-based Architectures

Model-based architectures in solving inverse problems are motivated by classical optimization. When the forward operator $\mathcal{A}$ is precisely known, one can unroll the objective function in (1) into numbers of iterative steps, and learn the regularization update from data. One subclass of model-based IP solvers is called *loop unrolling* (LU) or *deep unfolding*, which has a fixed number of iterations. LU is first introduced to solve sparse coding Gregor & LeCun (2010); Chen et al. (2018), and later on extended to many image restoration problems Monga et al. (2020); Gilton et al. (2021a); Ongie et al. (2020); Zhao et al. (2023; 2024). Variations

of LU emerge according to different optimization algorithms, such as projected gradient descent Lohit et al. (2019), proximal gradient descent Hosseini et al. (2020), alternating direction method of multipliers (ADMM) Yang et al. (2020), and momentum gradient descent Chun et al. (2023). More works can be found in Monga et al. (2020).

The concept of unrolling optimization iterations is not restricted to a fixed number but is often limited by memory constraints during backpropagation. To address this, Gilton et al. (2021a) proposes to train the iterative updates of the reconstructions until convergence using a fixed point solver. Similarly, Cai et al. (2021) extends a loop unrolling to an infinite number of iterations for robust principal component analysis, achieved by layer-wise training the initial iterations and updating the additional penalty parameters for subsequent iterations. In our work, we demonstrate the proposed forward model matching technique using DEQ, known for its infinite-layer extension with consistent architecture across all iterations.

## 2.2 Methods for Addressing Model Mismatch

A linear inverse problem with forward model mismatch can be interpreted as a parameter tuning within a predetermined structure. In other words, given a matrix $\boldsymbol{A}$ of dimension $m \times n$, the entries contain errors. This happens in many imaging tasks, such as blind deblurring (Li et al., 2020; Dong et al., 2020; Zhang et al., 2022) and blind deconvolution (Krishnan et al., 2011). These methods are tailored to specific tasks. For example, Li et al. (2020) is recast from TV-regularization due to the sparse gradients in natural images, Dong et al. (2020) incorporates operations in Fourier space, which is hard to generalize to other inverse problem tasks. Their task-specific designs often involve numerous hyperparameters, leading to high sensitivity in reconstruction (Cho & Lee, 2009). For example, we observe the instability in training the blind deblurring algorithm in Li et al. (2020) to hyperparameters such as threshold value, parameter initialization, scale of proximity, etc. In contrast, we propose a simple but stable solution applicable to general IPs with only estimated forward models.

On the other hand, when initial and true forward models differ structurally, updating parameters in $\mathcal{A}_0$ often yields suboptimal solutions, common in many nonlinear inverse problems. Oftentimes, $\mathcal{A}_0$ is the best approximation within this structure. This motivates the use of a more complex structure, like a neural network, to approximate the true model. In nonlinear applications, reconstructions with erroneous model estimates are typically tackled in two steps. First, training a neural network to learn the true forward model (Candiani et al., 2021; Hauptmann et al., 2018; Koponen et al., 2021; Liu et al., 2013; Lunz et al., 2021). Then, solving for the reconstruction using a predetermined regularizer like TV-norm (Candiani et al., 2021) and pseudo-Huber function Lunz et al. (2021), or learning a regularizer using the pre-trained forward model (Koponen et al., 2021). These approaches assume either a "perfect" learned forward model Koponen et al. (2021), or errors can be corrected via the learned regularizer Hauptmann et al. (2018) which is less interpretable. In this work, we propose to use an untrained neural network to match the data consistency, which allows updating the forward model and the reconstruction in one step adaptively for each instance.

# 3 Background

## 3.1 Loop Unrolling Methods

The traditional approach to solving inverse problems is by formulating them as an optimization problem of the form in (2). A natural approach to solving (2) is via a *proximal gradient descent* algorithm, which can be applied even when the regularizer $r$ is not differentiable, as is often the case (Parikh & Boyd, 2014). The resulting update involves first taking a gradient step (with a fixed step size $\eta$) that aims to minimize the data-fidelity term in (2). This is then followed by a proximal step, resulting in the iteration (Parikh & Boyd, 2014):

$$\boldsymbol{x}_{k+1} = \mathrm{prox}_{\gamma,r}(\boldsymbol{x}_k + \eta\mathcal{A}^*(\boldsymbol{y} - \mathcal{A}(\boldsymbol{x}_k))), \tag{3}$$

where $\mathcal{A}^*$ denotes the adjoint operator of $\mathcal{A}$, $\gamma > 0$ is again the parameter that controls the weight of the regularizer, and the proximal operator of a function $g$ is defined as:

$$\mathrm{prox}_g(\boldsymbol{x}) = \arg\min_{\boldsymbol{z}} \frac{1}{2}\|\boldsymbol{x} - \boldsymbol{z}\|_2^2 + g(\boldsymbol{z}). \tag{4}$$

As we can see, the choice of regularizer manifests itself entirely through the proximal operator. The LU algorithm essentially keeps the update in (3) but replaces the proximal operator with a neural network, and

limits the algorithm to a finite number of iterations $K$. The final output $\boldsymbol{x}_K$ is compared with the ground truth $\boldsymbol{x}$ and the network parameters are updated accordingly through *end-to-end* training.

By structuring the network in a way that mirrors proximal gradient descent, and taking advantage of an accurate descent direction derived from the knowledge of $\mathcal{A}$, the learned portion of the network can be interpreted as the proximal operator of a learned regularizer that enforces desired signal structures. However, when the forward model is inexact or only approximately known, the gradient update in (3) can introduce errors. Since LU is trained end-to-end, the error will manifest itself in the learned proximal operators. As a result, the neural network used in LU will no longer act as a pure proximal operator since it must both enforce signal structure as well as compensate for the errors in $\mathcal{A}$, potentially becoming less effective and interpretable.

## 3.2 Deep Equilibrium Models

*Deep equilibrium models* (DEQ) form a class of implicit neural networks, where the output is the fixed point solution of a neural network block Bai et al. (2019). For an initial input $\boldsymbol{x}^{(0)}$, and a neural network $\phi_\theta$, a DEQ repeatedly applies $\phi_\theta$,

$$\boldsymbol{x}^{(k+1)} = \phi_\theta(\boldsymbol{x}^{(0)}, \boldsymbol{x}^{(k)}),$$

until convergence to an equilibrium solution. The forward pass can be performed using any efficient root-finding algorithm. The backward pass, instead of saving all computational graphs to do backpropagation in one pass (also known as end-to-end training), can be performed by either direct computation of the vector-Jacobian product (Bai et al., 2019) or using the "Jacobian-free" strategy proposed in Fung et al..

Later work in Gilton et al. (2021a) extends LU using a DEQ architecture, so instead of having a pre-determined fixed number of iterations, this work allows a potentially high number of iterations until a fixed-point solution is reached. Using DEQ shows a higher reconstruction quality, lower memory usage, and observe consistent improvement in reconstruction.

## 3.3 Half-Quadratic Splitting (HQS)

Another key tool that we will rely on in our approach is *variable splitting*. Variable splitting is an iterative optimization method that solves problems where the objective function is a sum of multiple components (Geman & Yang, 1995; Nikolova & Ng, 2005; Bergmann et al., 2015; Hurault et al., 2022; Yang & Wang, 2017). It works by introducing an auxiliary variable and iteratively optimizing the objective function with respect to each variable while fixing the others. For example, by introducing an auxiliary variable $\boldsymbol{z}$, we can re-express the optimization problem in (2) as

$$\min_{\boldsymbol{x}, \boldsymbol{z}} \ \frac{1}{2}\|\boldsymbol{y} - \mathcal{A}(\boldsymbol{x})\|_2^2 + \gamma r(\boldsymbol{z}), \text{ s.t. } \boldsymbol{x} = \boldsymbol{z}.$$

The solution can be approximated using the following unconstrained problem,

$$\min_{\boldsymbol{x}, \boldsymbol{z}} \ \frac{1}{2}\|\boldsymbol{y} - \mathcal{A}(\boldsymbol{x})\|_2^2 + \gamma r(\boldsymbol{z}) + \frac{\mu}{2}\|\boldsymbol{x} - \boldsymbol{z}\|_2^2,$$

where $\mu \geq 0$ is a tuning parameter. This optimization can be solved by iteratively updating $\boldsymbol{x}_k$ and $\boldsymbol{z}_k$ until convergence:

$$\boldsymbol{x}_{k+1} = \arg\min_{\boldsymbol{x}} \ \frac{1}{2}\|\boldsymbol{y} - \mathcal{A}(\boldsymbol{x})\|_2^2 + \frac{\mu}{2}\|\boldsymbol{x} - \boldsymbol{z}_k\|_2^2,$$

$$\boldsymbol{z}_{k+1} = \arg\min_{\boldsymbol{z}} \ \gamma r(\boldsymbol{z}) + \frac{\mu}{2}\|\boldsymbol{z} - \boldsymbol{x}_{k+1}\|_2^2.$$

# 4 Proposed Method I: $\mathcal{A}$-adaptive Loop Unrolling Architecture

The discussion above has assumed exact knowledge of $\mathcal{A}$. Here we now suppose that we have an initial guess of the forward model $\mathcal{A}_0$ and consider a neural network $f_\theta$ that learns the measurement residual due to model misfit given the signal $\boldsymbol{x}$ and $\mathcal{A}_0$, i.e.,

$$\boldsymbol{y} = \mathcal{A}(\boldsymbol{x}) + \boldsymbol{\epsilon} = \mathcal{A}_0(\boldsymbol{x}) + f_\theta(\boldsymbol{x}, \mathcal{A}_0) + \boldsymbol{\epsilon}.$$

We assume that $\mathcal{A}_0$ is a useful estimate of the true forward model, in other words, $\mathcal{A}(\boldsymbol{x})$ and $\mathcal{A}_0(\boldsymbol{x})$ are close. We can then express the optimization problem in (2) as

$$\min_{\boldsymbol{x},\theta} \ \frac{1}{2}\|\boldsymbol{y} - \mathcal{A}_0(\boldsymbol{x}) - f_\theta(\boldsymbol{x}, \mathcal{A}_0)\|_2^2 + \gamma r(\boldsymbol{x}) + \tau\|f_\theta(\boldsymbol{x}, \mathcal{A}_0)\|_2^2. \tag{5}$$

Introducing an auxiliary variable $\boldsymbol{z}$, the solution of (5) is equivalent to solving

$$\min_{\boldsymbol{x},\boldsymbol{z},\theta} \ \frac{1}{2}\|\boldsymbol{y} - \mathcal{A}_0(\boldsymbol{z}) - f_\theta(\boldsymbol{z}, \mathcal{A}_0)\|_2^2 + \gamma r(\boldsymbol{x}) + \tau\|f_\theta(\boldsymbol{z}, \mathcal{A}_0)\|_2^2 + \lambda\|\boldsymbol{x} - \boldsymbol{z}\|_2^2. \tag{6}$$

Similar to the HQS updates, we first initialize $\boldsymbol{x}_0$, $\boldsymbol{z}_0$, and $\theta_0$. We then update each variable in the objective function by keeping other variables fixed. For $k = 1, 2, \ldots, K$, we have

$$\boldsymbol{z}_{k+1} = \arg\min_{z} \ \frac{1}{2}\|\boldsymbol{y} - \mathcal{A}_0(\boldsymbol{z}) - f_{\theta_k}(\boldsymbol{z}, \mathcal{A}_0)\|_2^2 + \tau\|f_{\theta_k}(\boldsymbol{z}, \mathcal{A}_0)\|_2^2 + \lambda\|\boldsymbol{x}_k - \boldsymbol{z}\|_2^2,$$

$$\theta_{k+1} = \arg\min_{\theta} \ \frac{1}{2}\|\boldsymbol{y} - \mathcal{A}_0(\boldsymbol{z}_{k+1}) - f_\theta(\boldsymbol{z}_{k+1}, \mathcal{A}_0)\|_2^2 + \tau\|f_\theta(\boldsymbol{z}_{k+1}, \mathcal{A}_0)\|_2^2, \tag{7}$$

$$\boldsymbol{x}_{k+1} = \text{prox}_{\frac{\gamma}{2\lambda}, r}(\boldsymbol{x}_k - \eta(\boldsymbol{x}_k - \boldsymbol{z}_{k+1})).$$

It should be noted that the update on $\boldsymbol{z}$ no longer has a closed-form solution due to the nonlinearity of $f_\theta$. However, this can be efficiently computed using Autograd in Pytorch (Paszke et al., 2017) or other differentiation computing algorithms. Meanwhile, the update on $\theta$ follows the regular backpropagation for network parameters. We can then replace the proximal operator with a neural network, and the update on $\boldsymbol{x}$ connects all the components to form an LU network as illustrated in Fig. 2. The parameters in the proximal network, are learned through end-to-end training, where $\eta$ is a trainable step size.

While we refer to $f_\theta$ as a network in the context of learning forward model mismatch, it differs from classical training approaches where weights remain fixed during evaluation. In our framework, the parameters $\theta$ in $f$ are updated both during training and evaluation for each instance. The objective is to iteratively align the data fidelity term for a specific measurement $\boldsymbol{y}$ through a nonlinear estimation $f$ with the minimal $\ell_2$-norm. Our proposed technique involves using untrained (random) weights $\theta_0$ at the initial iteration and adjusting $\theta_k$ based on the loss outlined in the second line of (7). The concept of employing an untrained neural network is reminiscent of Deep Image Prior (Ulyanov et al., 2018), which initializes a reconstruction network randomly, treating the weights as an implicit prior for the reconstruction. In our approach, however, we suggest using an untrained neural network as an integral part of the reconstruction process to address model mismatch while still being capable of learning regularization updates from all training data. Subsequent experiments demonstrate that during evaluation, the residual network $f$ can be initialized with various untrained weights, resulting in different performance levels.

Note that the proposed iterative updates look reminiscent of the alternating direction method of multipliers (ADMM), like Yang et al. (2016). When the forward model is precisely known, ADMM aims to minimize the Lagrangian of the objective function with an additional auxiliary variable. The proposed algorithm uses HQS as a simpler method to split variables in solving an optimization problem, but the algorithm could easily be generalized to other solvers such as ADMM.

## 5 Proposed Method II: $\mathcal{A}$-adaptive Deep Equilibrium Architecture

Loop unrolling algorithms typically restrict the number of iterations to a small value during end-to-end training, primarily to alleviate the high memory costs associated with processing a large number of steps. In such cases, the focus often leans more toward managing computational resources rather than achieving convergence. However, when it comes to another well-known model-based architecture the deep equilibrium model (DEQ) extension, the situation differs. The DEQ extension seeks a fixed-point solution, which requires careful consideration of convergence to ensure the model's effectiveness in finding the desired equilibrium. In this section, we first prove the convergence of the proposed iterative updates in (7) under mild conditions and thus propose an extension of $\mathcal{A}$-adaptive LU through the integration of a deep equilibrium model, namely $\mathcal{A}$-adaptive DEQ.

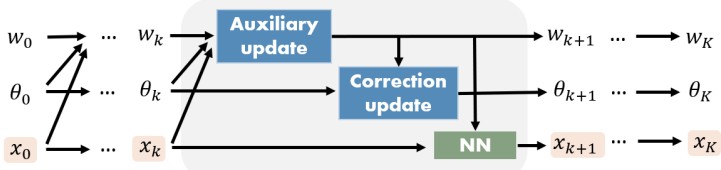

Figure 2: Illustration of the $k^{th}$ iteration of an $\mathcal{A}$-adaptive LU network. $\boldsymbol{x}_0$ is fed into the network, the auxiliary update and the correction update corresponding to updates in $\theta$ and $\boldsymbol{z}$ respectively, and the proximal network in green is updated using end-to-end training. The final output contains the parameters for estimating the function mismatch and the reconstruction estimate.

**Proposition 5.1.** *Assume $r$ is convex, and use the fact that $\|\boldsymbol{x} - \boldsymbol{z}\|_2^2$ is $L-$smooth with respect to $\boldsymbol{x}$, in another word let $f(\boldsymbol{x}) = \|\boldsymbol{x} - \boldsymbol{z}\|_2^2$, $\|\nabla f(\boldsymbol{x}_1) - \nabla f(\boldsymbol{x}_2)\|_2 \leq L\|\boldsymbol{x}_1 - \boldsymbol{x}_2\|_2$ for some $L > 0$. The algorithm in (7) converges when the step size $\eta_k = 1/L$ for all $k \geq 1$.*

*Proof.* Let $J(\boldsymbol{z}, \theta, \boldsymbol{x}) = \frac{1}{2}\|\boldsymbol{y} - \mathcal{A}_0(\boldsymbol{z}) - f_\theta(\boldsymbol{z}, \mathcal{A}_0)\|_2^2 + \gamma r(\boldsymbol{x}) + \tau\|f_\theta(\boldsymbol{z}, \mathcal{A}_0)\|_2^2 + \lambda\|\boldsymbol{x} - \boldsymbol{z}\|_2^2$. For a fixed $\boldsymbol{x}_k$ and $\theta_k$, $J(\boldsymbol{z}_{k+1}, \theta_k, \boldsymbol{x}_k) \leq J(\boldsymbol{z}_k, \theta_k, \boldsymbol{x}_k)$ because we are minimizing over $\boldsymbol{z}$. Similarly, given fixed $\boldsymbol{z}_{k+1}$ and $\boldsymbol{x}_k$, $J(\boldsymbol{z}_{k+1}, \theta_{k+1}, \boldsymbol{x}_k) \leq J(\boldsymbol{z}_{k+1}, \theta_k, \boldsymbol{x}_k)$ when minimizing over $\theta$. When fixing $\theta_{k+1}$ and $\boldsymbol{z}_{k+1}$, from the convergence result of the proximal gradient method (Parikh & Boyd, 2014), $J(\boldsymbol{z}_{k+1}, \theta_{k+1}, \boldsymbol{x}_{k+1}) \leq J(\boldsymbol{z}_{k+1}, \theta_{k+1}, \boldsymbol{x}_k) - \frac{1}{2L}\|L(\boldsymbol{x}_k - \boldsymbol{x}_{k+1})\|_2^2 \leq J(\boldsymbol{z}_{k+1}, \theta_{k+1}, \boldsymbol{x}_k)$. Therefore, the value of the objective function decreases for updates of $\boldsymbol{z}_k, \theta_k, \boldsymbol{x}_k$ at all $k$. In addition, because this value is also bounded below, the algorithm converges. $\square$

In contrast to employing a fixed number of iterations $K$ in $\mathcal{A}$-adaptive LU, as outlined in equation (7), $\mathcal{A}$-adaptive DEQ allows $\boldsymbol{x}_k$ to converge to a fixed point solution. This convergence is achieved over a potentially large number of iterations, motivated by the convergence property established in Proposition 5.1.

We denote $F_{\rho,\eta}$ as the function representing one iterative update of variables $\boldsymbol{z}_k$, $\theta_k$ and $\boldsymbol{x}_k$ in (7), where $\rho$ parameterizes the proximal neural network and $\eta$ denotes the trainable step size. According to Proposition 5.1, for a sufficiently large number of iterations $k_0$, a fixed point solution is reached for all $k > k_0$,

$$\boldsymbol{z}_k, \theta_k, \boldsymbol{x}_k = F_{\rho,\eta}(\boldsymbol{z}_k, \theta_k, \boldsymbol{x}_k).$$

$\mathcal{A}$-adaptive DEQ shares the same architecture as $\mathcal{A}$-adaptive LU, where $f$ is still an untrained neural network and the proximal operator is replaced using a neural network, but it is trained differently. Following the training scheme introduced in Gilton et al. (2021a), the forward pass of $\mathcal{A}$-adaptive DEQ solves for a fixed point solution, which can be efficiently achieved using Anderson acceleration (Walker & Ni, 2011). Whereas in the backward pass, we adopted the "Jacobian-free" backpropagation strategy stated in Fung et al.. We show in the experiments that the $\mathcal{A}$-adaptive DEQ performs even better than $\mathcal{A}$-adaptive LU due to a higher number of effective iterations. In addition, we verify its convergence during evaluation.

## 6 Experiments and Discussion

In this section, we demonstrate the efficacy of the proposed $\mathcal{A}$-adaptive LU algorithm and show an improvement in reconstruction by learning a more accurate forward model compared to the following baseline methods: 1) a neural network that maps from the initial reconstruction $\boldsymbol{x}_0$ to the ground-truth $\boldsymbol{x}$, referred to as a direct inverse mapping, which is independent of the forward operator, and 2) a robust LU that is trained with inexact $\mathcal{A}_0$'s from the initial $\boldsymbol{x}_0$. Notice that $\boldsymbol{x}_0$ is initialized with $\mathcal{A}_0^*\boldsymbol{y}$ when $\boldsymbol{x}$ and $\boldsymbol{y}$ are from different spaces or $\boldsymbol{x}_0 = \boldsymbol{y}$ if they are from the same space. We evaluate the algorithms on linear IP tasks (image blind deblurring and seismic blind deconvolution), and a nonlinear IP task (landscape defogging). For each task, the direct inverse network and the proximal network in LU and the proposed methods share the same architecture.

Notice that in linear IPs, the forward model is defined by a matrix in a specific dimension, so the true forward model can be determined by either solving for the least squares solution or by using a gradient

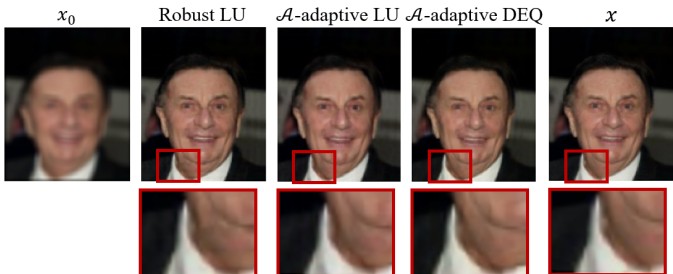

Figure 3: Comparing the deblurring results using robust LU, $\mathcal{A}$-adaptive LU and $\mathcal{A}$-adaptive DEQ to the ground truth, where $\boldsymbol{x}_0$ is the initial blurry images and $\boldsymbol{x}$ is the ground truth. The bottom row shows the zoomed regions in red boxes. The proposed methods generate sharper edges.

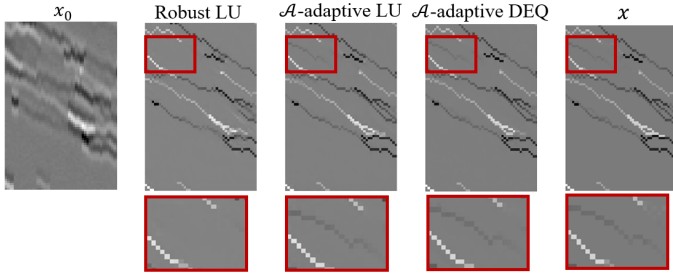

Figure 4: Comparing the deconvolution results using robust LU, $\mathcal{A}$-adaptive LU and $\mathcal{A}$-adaptive DEQ to the ground truth. The bottom row shows the zoomed regions in red boxes. A seismic layer in the red box is missing in reconstruction using the baseline robust LU method.

descent update inside the loop unrolling method. Our experiments first show the proposed methods work in linear cases, then more interestingly, when no precise formulation of the true forward model is known in nonlinear IPs (infeasible to solve directly), our proposed method shows its effectiveness in reconstruction.

**Image Blind Deblurring** In this problem, we aim to remove the blur from images $\boldsymbol{y}$ when a small amount of noise is present. The forward model is defined by a Gaussian blur kernel where we have inaccurate knowledge of the variance and size of the kernel. To validate this approach, experiments are conducted on the CelebA dataset (Liu et al., 2015), which was resized to $3 \times 120 \times 100$. The blurry images are generated with different Gaussian kernels for each pair of data $(\boldsymbol{x}_i, \boldsymbol{y}_i, \boldsymbol{A}_{0,i})$.

**Seismic Blind Deconvolution** Here an acoustic wave generated by a vibroseis truck on the surface of the earth propagates through the earth's layers. The reflected signals are collected by the geophones and stored as raw measurements. The measurements $\boldsymbol{y}$ are obtained in the form of (1). However, the frequencies of the acoustic wave are often inaccurately recorded due to the limited resolution and noise in the measurement process (Zabihi Naeini & Sams, 2017), which leads to an inaccurate estimate of the wavelet in the forward model. The forward process can be viewed as a convolution between the acoustic wave and the layer reflectivity $\boldsymbol{x}$. Therefore, the goal of seismic deconvolution is to reconstruct the layer reflectivities $\boldsymbol{x}$, with the inaccurate estimate of the wavelet in the forward model taken into account. The true model for each data pair $(\boldsymbol{x}_i, \boldsymbol{y}_i, \boldsymbol{A}_{0,i})$ can be expressed as follows,

$$\boldsymbol{y}_i = \boldsymbol{A}_{0,i}\boldsymbol{x}_i + f_\theta(\boldsymbol{x}_i, \boldsymbol{w}_i) + \boldsymbol{\epsilon}_i,$$

where $\boldsymbol{A}_{0,i}$ is a Toeplitz matrix derived from the inexact source wavelet $\boldsymbol{w}_i$. The measurement is simulated by applying an inaccurate wavelet with small additive Gaussian noise to the forward model. Notice that noise added to the true model may result in artifacts due to an extra magnification factor applied by $\boldsymbol{x}_i$, which is distinct from the measurement noise $\boldsymbol{\epsilon}_i$. The data is generated following the procedure in Iqbal et al. (2019).

**Landscape Image Defogging** We also demonstrate the proposed algorithms with a nonlinear IP. Consider the problem of removing the effects of fog, haze or mist from an image $\boldsymbol{y}$ to restore its true appearance

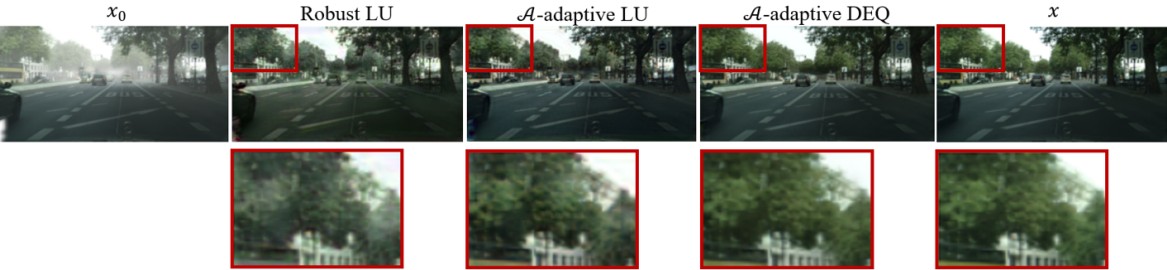

Figure 5: Comparing the defogging results using robust LU, $\mathcal{A}$-adaptive LU and $\mathcal{A}$-adaptive DEQ to the ground truth. The bottom row shows the zoomed regions in red boxes. The proposed methods reproduce cleaner images.

$\boldsymbol{x}$. Examples of $\boldsymbol{y}$ and $\boldsymbol{x}$ are illustrated in Fig. 5. The degradation process can be modeled as follows,

$$\boldsymbol{y} = \boldsymbol{x} \odot \boldsymbol{T}(\boldsymbol{x}) + \boldsymbol{L}(\boldsymbol{x}) \odot (\boldsymbol{I} - \boldsymbol{T}(\boldsymbol{x})) + \boldsymbol{\epsilon}, \tag{8}$$

where $\boldsymbol{T} \in [0,1]^{3 \times m \times n}$ is an unknown transmission map as a function of the fog-free image $\boldsymbol{x}$, $m$ and $n$ denote the width and height of the image. $\boldsymbol{T}(\boldsymbol{x})$ is then applied pointwise ($\odot$) to $\boldsymbol{x}$. $\boldsymbol{L}(\boldsymbol{x})$ is the unknown atmospheric light profile (Tufail et al., 2019). The goal is to estimate the model mismatch and recover the clean image. When the transition map is a matrix with all ones, the initial forward model is simplified to identity mapping pixel-wise. Since $\boldsymbol{T}$ and $\boldsymbol{L}$ are unknown functions with respect to $\boldsymbol{x}$, we use two separate networks to learn transmission mapping and the light profile respectively. This example shows the flexibility of our proposed methods. Instead of having one single network $f_\theta$ that learns the residual, if we know the explicit formulation of the true model, we can customize the residual network in a more structured way. To evaluate the proposed approaches, the CityScapes - Depth and Segmentation dataset given in Cordts et al. (2016) is used. This dataset provides depth maps, as well as foggy and clean views of urban street scenes.

## 6.1 Reconstruction Results

Table 1 compares the average and the standard deviation of the testing peak signal-to-noise ratio (PSNR) in dB and Structural Similarity Index Measure (SSIM) of the reconstructions. The neural network learns a direct inverse regardless of $\mathcal{A}_{0,i}$, which is the worst in reconstruction. It is notable that, even with some degree of inaccuracy in $\mathcal{A}$, the robust LU still outperforms the direct inverse mapping. This is because end-to-end training enables LU to fix the gradient-step errors resulting from the model mismatch to some extent, thereby maintaining reasonably higher-quality results. Our results align with the finding in Krainovic et al. (2023) that the robust LU trains an inverse mapping that is less sensitive to perturbations in the $\boldsymbol{y}$-space, by seeing inaccuracies in the forward model at train time. However, in the next section, we will discuss the intermediate reconstruction ineffectiveness of robust LU compared to the proposed methods.

The proposed methods make an improvement by actively adjusting for the model mismatch for each data pair. Instead of having the neural network learn both the proximal step and the model correction as in a black box, the proposed methods separate the tasks and each part of the network is now learning a clearer and easier task, resulting in a substantial performance gain over both the robust LU and the direct inverse mapping in all tasks. Furthermore, $\mathcal{A}$-adaptive DEQ allows for a higher number of iterations than $\mathcal{A}$-adaptive LU, resulting in superior performance in both PSNR and SSIM. Notice that the $\mathcal{A}$-adaptive LU tends to have a smaller standard deviation than $\mathcal{A}$-adaptive DEQ because the latter varies in number of iterations.

Apart from the quantitative evaluation, we offer a visual comparison of the reconstruction results using the robust LU and the proposed methods for the presented tasks in Figs. 3, 4, and 5. In blind image deblurring, Fig. 3 shows that the proposed methods preserve a sharper jawline. In seismic blind deconvolution, the baseline method (Robust LU) fails to reconstruct a seismic layer as highlighted in red boxes in Fig. 4, whereas both proposed methods successfully capture the layer. Furthermore, in the landscape defogging example, Fig. 5 demonstrates the proposed methods remove more artifacts while preserving more detailed information.

Table 1: Average and standard deviation of testing PSNR and SSIM for direct inverse mapping, robust LU, $\mathcal{A}$-adaptive LU, and $\mathcal{A}$-adaptive DEQ. The best performances for each task are in bold and the second best performances are underlined.

|  |  | Deblur. | Deconv. | Defog. |
|---|---|---|---|---|
| Direct Inverse | PSNR | $24.002 \pm 0.101$ | $22.816 \pm 0.112$ | $21.802 \pm 1.706$ |
|  | SSIM | $0.7689 \pm 0.0007$ | $0.7421 \pm 0.0041$ | $0.8738 \pm 0.0347$ |
| Robust LU | PSNR | $34.380 \pm 0.437$ | $24.785 \pm 0.836$ | $29.645 \pm 1.939$ |
|  | SSIM | $0.9429 \pm 0.0014$ | $0.8407 \pm 0.0214$ | $0.9563 \pm 0.0164$ |
| $\mathcal{A}$-adaptive LU (proposed meth. I) | PSNR | $\underline{36.621} \pm 0.330$ | $\underline{27.271} \pm 0.787$ | $\underline{31.112} \pm 1.854$ |
|  | SSIM | $\underline{0.9731} \pm 0.0010$ | $\underline{0.8960} \pm 0.0157$ | $\underline{0.9661} \pm 0.0133$ |
| $\mathcal{A}$-adaptive DEQ (proposed meth. II) | PSNR | $\mathbf{37.809} \pm 0.532$ | $\mathbf{28.236} \pm 0.321$ | $\mathbf{32.149} \pm 2.069$ |
|  | SSIM | $\mathbf{0.9774} \pm 0.0041$ | $\mathbf{0.9082} \pm 0.0201$ | $\mathbf{0.9723} \pm 0.0124$ |

## 6.2 Effectiveness of Reconstruction

We also explore the effectiveness of reconstruction by comparing the intermediate reconstruction $\boldsymbol{x}_k$'s in the robust LU and $\mathcal{A}$-adaptive LU in Fig. 6. The average testing Mean-Squared Error (MSE) between $\boldsymbol{x}_k$'s and the ground truth $\boldsymbol{x}$ are recorded for all $k = 1, ..., 5$. While LU is trained to be robust to mismatch in $\mathcal{A}$'s, the intermediate MSE of the robust LU remains high until the final iteration. In fact, at iteration 3 in the image deblurring task and iteration 2 and 4 in the landscape defogging task, we observe noticeable MSE increments in the robust LU (solid blue curve). It indicates that the robust LU lacks the capacity to learn how to effectively correct the errors in a single iteration. Because the proximal network in the robust LU aims to perform both the proximal step and the model mismatch, it learns a less interpretable mapping from the erroneous gradient step to the true $\boldsymbol{x}$. In contrast, $\mathcal{A}$-adaptive LU learns model mismatch and the proximal step separately. It thus allows effective error correction and quality improvement in every iteration, resulting in a consistent MSE decrement.

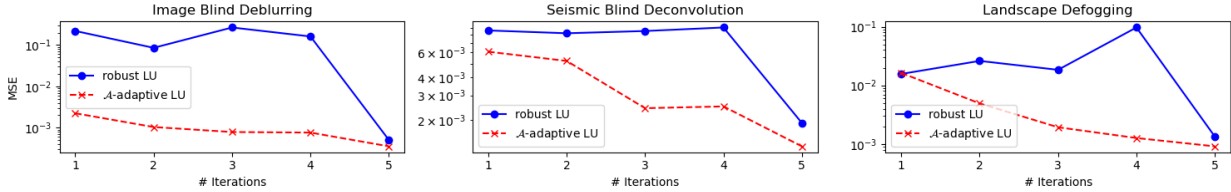

Figure 6: Average testing MSE of the intermediate reconstructions for the 5-iteration robust LU and the proposed $\mathcal{A}$-adaptive LU.

## 6.3 Intermediate Reconstructions of $\mathcal{A}$-adaptive DEQ

We show the intermediate reconstructions of the DEQ variant of the proposed method in Fig. 7. The proposed network is trained with a maximum of 30 iterations or when a fixed point solution is reached, whereas when evaluating $\mathcal{A}$-adaptive DEQ with more iterations (i.e., 100), the reconstruction error remains low. This result aligns with the finding in Gilton et al. (2021a) where the reconstruction quality remains high after reaching the fixed point, particularly when the forward model is precisely known.

## 6.4 Robustness of $\mathcal{A}$-adaptive LU with More Iterations

In this section, we empirically assess the stability of our proposed $\mathcal{A}$-adaptive LU variant by evaluating it with a greater number of unrolling iterations than those employed during training. The result can be found in 8. Across all three tasks, a noticeable Mean Squared Error (MSE) gap emerges between the two curves. The solid blue curve, representing robust LU, exhibits a drop in MSE at iteration 5, which corresponds to the target output during training. However, as the number of iterations exceeds the max iterations of training (in this set of experiments $k > 5$), the MSE rises significantly. Notice that although the MSE

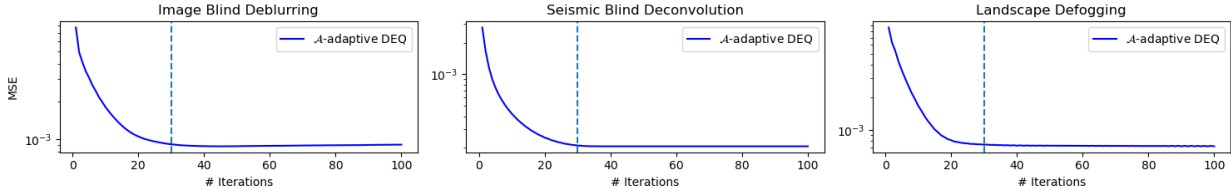

Figure 7: Average testing MSE of the intermediate reconstructions for the proposed $\mathcal{A}$-adaptive DEQ. A max of 30 iterations is used in training, this figure depicts the extended evaluation result when allowing a max of 100 iterations.

at iteration 10 is relatively low for the landscape defogging task, the reconstruction is not stable and less predictable. In contrast, the dashed red curve ($\mathcal{A}$-adaptive LU) oscillates around the lowest MSE. This observation highlights the enhanced robustness of the proposed $\mathcal{A}$-adaptive LU when subjected to a higher number of iterations during evaluation.

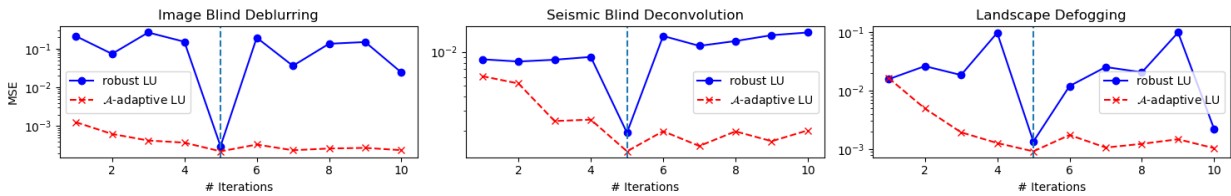

Figure 8: Average testing MSE of intermediate reconstructions using the robust LU and $\mathcal{A}$-adaptive LU with 10 unrolling iterations, while trained on 5 unrolling iterations.

## 6.5 Random Initialization of the Mismatch Network

The residual network $f_\theta$ is updated along the reconstruction in both training and evaluation for each data instance. Because the update of $\theta$ in (7) aims to minimize the data-fidelity term with consideration of the forward model mismatch in the smallest $\ell_2$-norm, the initialization of $\theta$ is less important. In this section, we show that $f_\theta$ can be initialized by an untrained neural network at even evaluation time, and still obtain the same level of performance in both proposed architectures. We evaluate the PSNR and SSIM of reconstruction results from both proposed methods using a saved model $f_\theta$, and two commonly used neural network initialization methods: Kaiming uniform (He et al., 2015) and Xavier (Glorot & Bengio, 2010) initialization, as shown in Table 2.

Table 2: Average testing PSNR and SSIM for $\mathcal{A}$-adaptive LU and $\mathcal{A}$-adaptive DEQ using $f_\theta$ with the saved model, uniform random initialization, and Xavier initialization.

|  | PSNR/SSIM | $\mathcal{A}$-adaptive LU | $\mathcal{A}$-adaptive DEQ |
|---|---|---|---|
| | saved model | 36.621/0.9731 | 37.809/0.9774 |
| Deblurring | uniform | 36.604/0.9730 | 37.808/0.9774 |
| | Xavier | 36.606/0.9731 | 37.809/0.9773 |
| | saved model | 27.271/0.8960 | 28.236/0.9082 |
| Deconvolution | uniform | 27.268/0.8959 | 28.239/0.9084 |
| | Xavier | 27.272/0.8960 | 28.238/0.9084 |
| | saved model | 31.112/0.9661 | 32.149/0.9723 |
| Defogging | uniform | 31.277/0.9661 | 32.147/0.9722 |
| | Xavier | 31.276/0.9661 | 32.147/0.9723 |

## 6.6 Discussion of Evaluation Runtime

While the suggested methods demonstrate superior numerical and visual reconstruction qualities, they do incur a runtime tradeoff to be more accurate. Table 3 compares the average and standard deviation of runtime per batch during evaluation. The same batch sizes are maintained across all three methods for each task to ensure comparable computational overhead. The loop unrolling variant of the proposed method is not significantly slower than the robust LU, despite the $\mathcal{A}$-adaptive LU involving extra optimization steps within the reconstruction process compared to the one-step gradient update in the robust LU. However, the DEQ variant is noticeably slower due to its high number of iterations in solving for a fixed point solution. The $\mathcal{A}$-adaptive DEQ method also has higher variance because the number of iterations is not fixed due to the design of DEQs.

Table 3: Average and standard deviation of runtime in ms during evaluation for robust LU, $\mathcal{A}$-adaptive LU and $\mathcal{A}$-adaptive DEQ. The same batch size is enforced for all methods across each task.

| Tasks | Methods | Evaluation time (ms)/batch |
|---|---|---|
| Deblurring (batch size = 32) | robust LU | $80.26 \pm 0.52$ |
| | $\mathcal{A}$-adaptive LU | $253.06 \pm 0.82$ |
| | $\mathcal{A}$-adaptive DEQ | $2219.53 \pm 15.89$ |
| Deconvolution (batch size = 32) | robust LU | $81.14 \pm 0.67$ |
| | $\mathcal{A}$-adaptive LU | $256.94 \pm 0.85$ |
| | $\mathcal{A}$-adaptive DEQ | $2249.60 \pm 28.37$ |
| Defogging (batch size = 8) | robust LU | $112.91 \pm 1.03$ |
| | $\mathcal{A}$-adaptive LU | $370.93 \pm 1.14$ |
| | $\mathcal{A}$-adaptive DEQ | $2433.66 \pm 15.79$ |

## 7 Conclusion

While the model-based machine learning IP solvers are powerful, they demand precise knowledge of the forward models, which can be impractical in many applications. One way to handle the model mismatch is to train a model with inaccurate forward models, which will implicitly correct the errors due to end-to-end training and demonstrate reasonable performance. However, this falls short in terms of interpretability and efficacy in reconstruction. To address this problem, we introduce a novel procedure to adapt for the forward model mismatches using untrained neural networks actively based on two well-known model-based machine learning architectures, denoted as $\mathcal{A}$-adaptive LU and $\mathcal{A}$-adaptive DEQ. Experimental results across both linear and nonlinear inverse problems showcase the effectiveness of the proposed methods in learning forward model updates. Consequently, they surpass the baseline methods significantly when trained as a robust reconstruction mapping to accommodate variations in $\mathcal{A}$. We further demonstrate the robustness of the proposed methods to random initialization of the residual block and a higher number of iterations during evaluation. Finally, we show the accuracy-runtime tradeoff in handling model mismatch using the proposed variants of the model-based architectures.

### Broader Impact Statement

This work adaptively addresses the errors in the forward model by optimizing the auxiliary variable and the untrained mismatch neural network within a model-based architecture. We show the flexibility of the proposed forward model matching module using two well-known architectures. However, the accuracy-runtime tradeoff still exists. Although the proposed $\mathcal{A}$-adaptive DEQ variant achieves the best performance for all tasks, when solving large-scale inverse problems, careful consideration of runtime is necessary. Algorithms such as momentum methods accelerate the convergence speed of an optimization problem. Investigating a more sophisticated design with accelerated algorithms offers an intriguing prospect for future work to expedite $\mathcal{A}$-adaptive DEQ.

### Acknowledgments

This work was supported by the Center for Energy and Geo Processing (CeGP) at Georgia Tech, the Deanship of Research Oversight and Coordination (DROC), King Fahd University of Petroleum and Minerals (KFUPM), under Project GTEC2013.

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

## A   Appendix: Training Details

The proximal networks for LU and $\mathcal{A}$-adaptive LU are kept the same for each task, with 5-layer DnCNN for image deblurring and 7-layer DnCNN for landscape defogging. In the forward mismatch network, $\mathcal{A}_0(\boldsymbol{x})$ combines with $\mathcal{A}_0$ features, processed through a 3-layer CNN. $\mathcal{A}_0$ features comprise an incorrect Gaussian kernel for deblurring and landscape depth profiles. Because the initialization of $f_\theta$ is less important, the parameters $\theta$ are not reinitialized for each training instance for better time efficiency. Both LU and $\mathcal{A}$-adaptive LU employ a maximum of $K = 5$ iterations. The hyper-parameters in $\mathcal{A}$-adaptive LU (6) are chosen via sensitivity analysis, where $\lambda = 0.01$, $\tau = 0.1$ for deblurring, and $\lambda = 0.0001$, $\tau = 0.1$ for defogging. Learning rates are set at $= 0.0001$ for updating $\boldsymbol{z}$ and network parameters. All networks are trained on NVIDIA RTX 3080 with 10GB of RAM.

## B   Appendix: Effect of $\mathcal{A}_0$ in Reconstruction

In addition, we show that our proposed methods are less sensitive to the quality of the initial forward model. We demonstrate the idea using the image deblurring task, where one has flexibility in choosing the initial forward model. Assuming the true forward model for an image deblurring problem has a Gaussian kernel size of 5 with a variance of 7, we consider initial forward models $\boldsymbol{A}_0$ constructed with Gaussian kernels of sizes 5, but variances of 1, 3, 5, and 7. Figure 9 illustrates the reconstruction results with different $\boldsymbol{A}_0$'s for each method. When $\boldsymbol{A}_0$ deviates from $\boldsymbol{A}$, robust LU exhibits noticeable artifacts (with $\sigma = 1$ and 3), while our proposed methods show minimal degradation in quality even with significant differences between $\boldsymbol{A}_0$ and $\boldsymbol{A}$. This is due to the adaptive nature of our methods in iteratively addressing model mismatches. Notice that $\sigma = 1$ and 3 are treated as extreme cases, which were never encountered in $\boldsymbol{A}$ and $\boldsymbol{A}_0$ during training. The proposed methods generalize better to unseen estimated forward models.

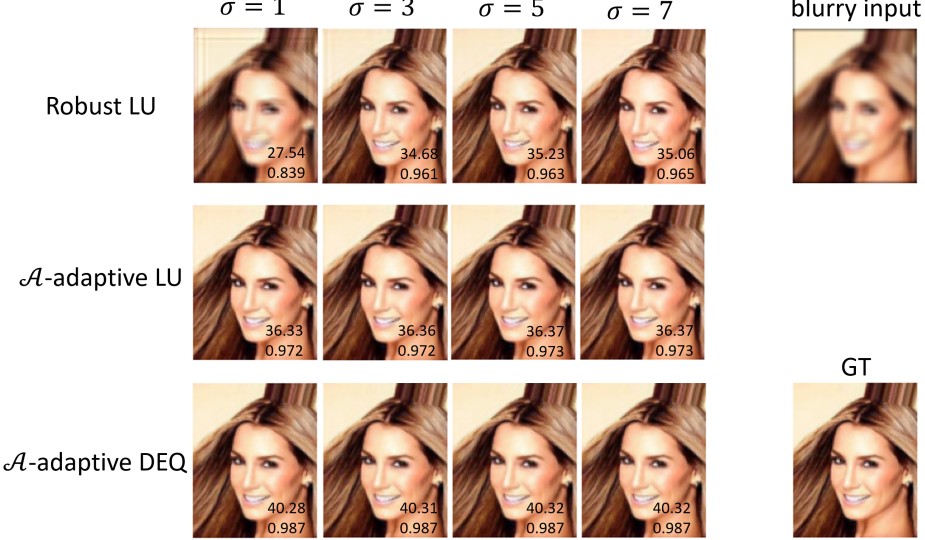

Figure 9: Visualizing the deblurred images using different $\boldsymbol{A}_0$, where $\sigma$ denotes the variance of the Gaussian kernel of the estimated forward model. The numbers in each image denote the PSNR and SSIM values.

