# OpenReview forum: "Solving Inverse Problems with Model Mismatch using Untrained Neural Networks within Model-based Architectures"
_TMLR — Accepted by TMLR_

### Review · Reviewer_43k2 · 2024-03-03

**Summary Of Contributions:**

The paper studies the regularized inverse problem with loop unrolling (LU) or deep equilibrium (DEQ) frames. It focuses on the case when there is small mismatching in the sensing matrix $\mathcal{A}$. Two methods (one for LU and one for DEQ) were proposed to adapt the mismatched $\mathcal{A}$ and find better solutions than the existing robust loop unrolling method.

**Audience:**

Yes

**Broader Impact Concerns:**

No Broader Impact Concerns.

**Claims And Evidence:**

Yes

**Requested Changes:**

See weakness.

**Strengths And Weaknesses:**

Strengths:

As far as I read, the proposed technique has no major flaws. The paper is well-written. There is a reasonable amount of numerical evidence to support the effectiveness of the proposed methods.

Weaknesses:

The image applications tackled by this paper may not attract much interest in the ML community as deep learning methods have extremely strong performance in those CV tasks. However, the base optimization problem may find other interesting applications.It would be interesting if the authors could list some more applications where a mismatched $\mathcal{A}$ is common.

The convergence theorem based on monotony objective value decreasing is a weak theoretical result. It would be interesting if the authors could show more useful theoretical insights, such as the convergence rate.

Some numerical experiments feature the $\mathcal{A}$-adaptive LU, some feature the $\mathcal{A}$-adaptive DEQ, and the rest feature both. How were those decisions made? In the case both $\mathcal{A}$-adaptive LU and $\mathcal{A}$-adaptive DEQ are compared, it seems $\mathcal{A}$-adaptive DEQ is also significantly better. Does it suggest that proposing only $\mathcal{A}$-adaptive DEQ is enough for this paper? Is $\mathcal{A}$-adaptive LU just a step stone here?

Some other deep unrolling methods are not limited to a finite number of iterations, but were missed in the discussion. For example, [R1] proposed an FNN-RNN-mixed unrolling which extended the LU model to infinitely many iterations. This mixed structure has the advantages of both LU and DEQ. The author may find it interesting.

[R1]  Learned Robust PCA: A Scalable Deep Unfolding Approach for High-Dimensional Outlier Detection. NeruIPS 2021

---

> ### Author Response · Authors · 2024-03-18
> **Response to reviewer 43k2**
>
> We appreciate the thoughtful review by this respected reviewer. The reviewer's concerns are addressed point-by-point as follows.
>
>
> > It would be interesting if the authors could list some more applications where a mismatched is common
>
> Besides the examples mentioned in the introduction, some other examples include sparse signal recovery when the sensing matrix is not known perfectly, seismic tomographic reconstruction when the source signal is corrupted, and computational tomography (CT) when the dynamic behavior of a human object is observed during the measurement period. We have also included this response to the revised manuscript in the first paragraph of the Introduction in Section 1.
>
> > It would be interesting if the authors could show more useful theoretical insights, such as the convergence rate.
>
> In this work, we propose to train an unfolded iterative inverse problem solver with the deep equilibrium model (DEQ), which is embedded in a fixed-point solver. Our theorem demonstrates the convergence of the proposed algorithm to justify the use of DEQ. The convergence rate may affect the runtime, which can be an interesting future work direction.
>
> > Does it suggest that proposing only $\mathcal{A}$-adaptive DEQ is enough for this paper? Is $\mathcal{A}$-adaptive LU just a step stone here?
>
> Loop unrolling and its DEQ extension are two famous model-based architectures for solving inverse problems, and we aim to show the proposed idea of using an untrained neural network to fit forward model mismatch can be applied to general model-based architectures. Moreover, although $\mathcal{A}$-adaptive DEQ shows higher performance than $\mathcal{A}$-adaptive LU, in section 6.7, we also demonstrate the DEQ variant is significantly slower. Depending on the task one aims to solve, there is flexibility in choosing either variant based on this accuracy and runtime tradeoff.
>
> > Some other deep unrolling methods are not limited to a finite number of iterations, but were missed in the discussion.
>
> We thank the reviewer for bringing up [1]. We acknowledge that there are other deep unrolling methods that allow an infinite number of iterations, such as FRMNN presented by [1], and DEQ presented by [2]. This work demonstrates the proposed forward model matching technique using DEQ, as it represents the most well-known infinite-layer extension with consistent architecture across all iterations. In contrast, FRMNN trains the initial $K$ unfolding iterations layer-wise, and for iterations $k \geq K$, it incorporates additional penalty parameters to the learned step size and the threshold in the network while keeping the weights fixed. We would like to emphasize that it is feasible to incorporate our proposed forward model matching technique into the architecture in [1]. Thanks for highlighting the relevance of these works, and we have added a comprehensive discussion of relevant methods to the revised manuscript. Improving runtime and memory usage will be left as future work and has been added to the Broader Impact section of the revised manuscript.
>
>
> [1] Cai et al (2021) Learned Robust PCA: A Scalable Deep Unfolding Approach for High-Dimensional Outlier Detection. NeruIPS 2021
>
> [2] Gilton et al (2021) Deep equilibrium architectures for inverse problems in imaging. IEEE Transactions on Computational Imaging

---

> > ### Comment · Reviewer_43k2 · 2024-04-08
> >
> > Thank you for your response, all my concerns have been addressed.

---

### Review · Reviewer_Wz4e · 2024-03-03

**Summary Of Contributions:**

In order to solve the forward model mismatch problem, this paper introduces a novel procedure to adapt for the forward model mismatches using untrained neural networks actively based on two well-known model-based machine learning architectures, denoted as A-adaptive loop unrolling (LU) and A-adaptive deep equilibrium model (DEQ). Theoretically, convergence of A-adaptive DEQ has been proved under mild conditions. Practically, experiments have been conducted to demonstrate a significant quality improvement in removing artifacts and preserving details within three distinct applications, covering both linear and nonlinear inverse problems. In addition, comprehensive analysis has been provided from the perspective of robustness, random initialization as well as evaluation runtime.

**Audience:**

Yes

**Broader Impact Concerns:**

The presented work innovatively tackles inaccuracies within the forward model by refining the auxiliary variable alongside an untrained neural network, specifically designed to handle discrepancies, all within a structured model-based framework. This approach demonstrates its versatility by integrating seamlessly with two established architecture types. Nevertheless, the balance between accuracy and computational time remains a challenge. The newly introduced \mathcal{A}-adaptive DEQ model variant outperforms existing solutions across various tasks. It is crucial to weigh the computational demands, especially for extensive inverse problem-solving scenarios, indicating that achieving higher performance levels may necessitate increased processing time. This careful balancing act underscores the need for a judicious evaluation of runtime implications in large-scale applications.

**Claims And Evidence:**

Yes

**Requested Changes:**

There are certain details I am concerned about. First of all, I am little bit confused about the f_{\theta} as a network in the context of learning forward model mismatch in both \mathcal{A}-adaptive loop unrolling architecture and \mathcal{A}-adaptive deep equilibrium architecture. It is said that the network f is untrainable however the parameters \theta in f are updated both during training and evaluation, which causes me a certain confusion on whether the network f is trainable or not. Probably I have a misunderstanding about that, could you please give me more details or specific example on it. Besides, I am also lost F_{\rho,\eta} in the section 4 Proposed Method II: \mathcal{A}-adaptive deep equilibrium architecture, and it is illustrated that \rho parameterizes the proximal neural network and \eta denotes the trainable step size, so do you mean the proximal neural network is trainable and how \rho parameterizes the neural network. Please provide more details about it.

In terms of the experiments for linear inverse problems, like the image blind deblurring, the forward model is defined by a Gaussian blur kernel with inaccurate knowledge of the variance and size of kernel. Is it possible to use the blur kernel in our real world, for example, like the non-linear kernel generated by recording the camera motion trajectories mentioned in [1]? It seems more meaningful to use the real blur kernel compared with the Gaussian blur one in order to valid the effectiveness of the proposed \mathcal{A}-adaptive LU and \mathcal{A}-adaptive DEQ.

[1] Y. Li, M. Tofighi, J. Geng, V. Monga and Y. C. Eldar, ``Efficient and interpretable deep blind image deblurring via algorithm unrolling’’, IEEE Transactions on Computational Imaging, vol. 6, 2020.

Moreover, it seems that no matter \mathcal{A}-adaptive LU and \mathcal{A}-adaptive DEQ, it is necessary to know the initial guess of the forward model \mathcal{A}_{0}. So how do you determine or what do you adopt the \mathcal{A}_{0}, especially in the defogging case?

Last but not least, I recommend the author to compare his proposed methods (\mathcal{A}-adaptive LU and \mathcal{A}-adaptive DEQ) with other latest methods for solving the inverse problems (IP), for example, (1) the Deep Unrolling for Blind Deblurring (DUBLID)[1], the one who unrolled the HQS algorithm with learnable parameters, (2) the Deep Convergent Unrolling for Non-blind deblurring (DECUN)[2] which also provides convergence similar to that of \mathcal{A}-adaptive DEQ, and (3) Momentum-Net [3] .etc

[1] Y. Li, M. Tofighi, J. Geng, V. Monga and Y. C. Eldar, ``Efficient and interpretable deep blind image deblurring via algorithm unrolling'', IEEE Transactions on Computational Imaging, vol. 6, 2020.
[2] Y. Zhao, Y. Li, H. Zhang, V. Monga and Y. C. Eldar, ``Deep, convergent, unrolled half-quadratic splitting for image deconvolution'', arXiv preprint arXiv:2402.12872.
[3] I. Y. Chun, Z. Huang, H. Lim, and J. A. Fessler, “Momentum-Net: Fast and convergent iterative neural network for inverse problems,” IEEE Trans. Pattern Anal. Mach. Intell., 45(5):4915–4931, Apr. 2023. doi: 10.1109/TPAMI.2020.3012955.

**Strengths And Weaknesses:**

Strengths: This paper proposes a simple but efficient architecture dubbed \mathcal{A}-adaptive LU and \mathcal{A}-adaptive DEQ taking advantage of residual network for solving the forward model mismatch problem. This paper is easy-understanding and straightforward, and it also provides a comprehensive analysis of the proposed model, including stability, random initialization as well as evaluation runtime.

Weaknesses: Some details on the model’s architecture should be provided in case of any confusion, and more experimental comparison with some state-of-art methods also need to be provided. The detailed concerns are listed in the next section.

---

> ### Author Response · Authors · 2024-03-18
> **Response to reviewer Wz4e (part 1)**
>
> We appreciate the thoughtful review by this respected reviewer. The reviewer's concerns are addressed point-by-point as follows.
>
> > Clarification of the untrained network $f_\theta$.
>
> the "untrained" neural network $f_\theta$ in our work means no pre-training is required during evaluation. The weights $\theta$ are not updated based on the training loss, but are updated for each testing input $y$ within one pass of a model-based reconstruction. In addition, we adopted the term "untrained" neural network to maintain consistency with a class of machine learning algorithms in the inverse problem literature that minimizes a loss function for a single data point. Some well-known untrained neural networks used in solving inverse problems include Deep Image Prior (DIP) [1] and deep decoder [2].
>
> Here are some detailed explanations, the network $f_\theta $ aims to reduce the forward model mismatch for a single measurement $y$. Unlike the classical neural network training scheme, which typically fits the curve based on the aggregate of all data in a dataset, $f_\theta$ minimizes the second equation in (7) by iteratively updating $\theta$ throughout the reconstruction for a single measurement $y$. Notice that no ground truth $x$ is required for updating $\theta$, where paired data $(x, y)$ are required to train a classical neural network. Thus we claim $f_\theta$ is untrained.
>
> > Is it possible to use the blur kernel in our real world?
>
> yes. While we demonstrate our work using a Gaussian kernel, our method imposes no restrictions on kernel types. Our methods can work with arbitrary blur kernels. Gaussian deblurring is also an important real-world problem when a camera loses focus. Our demonstration using the Gaussian kernel validates the use of the proposed approach for other kernels as well.
>
> > It is necessary to know the initial guess of the forward model $\mathcal{A}_0$. So how do you determine or what do you adopt the $\mathcal{A}_0$, especially in the defogging case?
>
> The vanilla model-based architectures such as loop unrolling and its DEQ extensions require precise knowledge of the forward model $\mathcal{A}$. Other model-independent IP solvers are less interpretable and often lead to degradation in performance.
> In this work, we made a relaxed assumption that only an approximation of the forward operator $\mathcal{A}_0$ is required. This approach is commonplace in practical applications, where simplified models are often employed to represent complex physical phenomena or to account for minor discrepancies in sensor locations relative to the mathematical model. The forward model can be estimated differently for each scenario, but how to estimate is out of the scope of this paper.
>
> To answer the question of initializing $\mathcal{A}_0$ in the defogging problem, we assume the forward model follows the degradation process in Eq. (8) with an unknown transmission map $T$ and light profile $L$. When the transition mapping $T$ is a matrix with all ones, the initial forward model is simplified to identity mapping pixel-wise. The choice of the initial model for the defogging problem is added and highlighted in Section 6 of the revised manuscript.

---

> ### Author Response · Authors · 2024-03-18
> **Response to reviewer Wz4e (part 2)**
>
> > How is the proposed work compared to Deep Unrolling for Blind Deblurring (DUBLID) [3], Deep Convergent Unrolling for Non-blind deblurring (DECUN) [4] and Momentum-Net [5].
>
> DUBLID [3] solves blind motion deblurring problems based on TV-regularization in the gradient domain, which assumes some form of prior of the ground-truth image such as assuming natural images tend to have sparse gradients or edges, and DUBLID learns a set of parameters to encourage sharp edges and smooth regions in the reconstructed image. In contrast, we proposed a simple framework that imposes no assumptions (i.e. sharp edges or smoothness) to the ground-truth image and is capable of solving forward model mismatches in general inverse problems.
>
> During our investigation, we encountered instability in training DUBLID within the given timeframe. DUBLID is tailored to solve deblurring problems, which involve numerous task-specific hyperparameters including the total number of layers, the scale of proximity list to previous solutions, soft threshold, learning rate, and initialization of trainable parameters. This training and reconstruction are sensitive to the choice of hyperparameters. [6] observed a similar instability to various hyperparameters in a different task-specific reconstruction algorithm. Indeed, the original DUBLID code utilizes gradient clipping to mitigate unstable training. In contrast, we proposed a general framework that entails fewer hyperparameters, thus our methods exhibit significantly improved stability during training. While these hyperparameters may influence the reconstruction quality, they have less impact on stability.
>
> In addition, DECUN [4] considers the non-blind deconvolution problem, which assumes knowing $\mathcal{A}$ precisely. [5] is designed to speed up convergence in an iterative neural network. They are out of the scope of our paper. It's worth mentioning that although [4] also used HQS, which is a widely-used variable splitting algorithm, our work aims to solve the forward model mismatch.
>
> All the detailed discussions have been added to the revised manuscript in Section 2 Related Works.
>
>
>
> [1] Ulyanov et al (2018) Deep image prior. CVPR 2018
>
> [2] Heckel et al (2019) Deep decoder: Concise image representations from untrained non-convolutional networks
>
> [3] Li et al (2020) Efficient and interpretable deep blind image deblurring via algorithm unrolling, IEEE Transactions on Computational Imaging, vol. 6, 2020.
>
> [4] Zhao et al (2024) Deep, convergent, Unrolled Half-quadratic Splitting for Image Deconvolution.
>
> [5] Chun et al (2023) Momentum-Net: Fast and convergent iterative neural network for inverse problems, IEEE Trans. Pattern Anal. Mach. Intell.
>
> [6] Cho et al (2009) Fast motion deblurring. ACM SIGGRAPH 2009.

---

> > ### Comment · Reviewer_Wz4e · 2024-03-23
> >
> > Thank you very much for your detailed response. Most of my concerns have been addressed, however, one reference ([4] Zhao et al (2024) Deep, convergent, Unrolled Half-quadratic Splitting for Image Deconvolution.) you have discussed in Section 2 Related Works have a short conference version, as follows:
> >
> > Zhao et al (2023) A convergent neural network for non-blind image deblurring, IEEE International Conference on Image Processing (ICIP).
> >
> > It had better give a comprehensive reference if you have discussed the work [4].

---

> > > ### Author Response · Authors · 2024-03-23
> > > **Response to reviewer Wz4e**
> > >
> > > Thank you for your acknowledgment. We are glad to hear that most of your concerns have been addressed. In addition, the ICIP version of the suggested work has also been added to the revised manuscript.

---

### Review · Reviewer_tfs5 · 2024-03-09

**Summary Of Contributions:**

The paper proposes two new methods for inverse problem solving based on loop unrolling (LU) and deep equilibrium model (DEQ).

**Audience:**

Yes

**Broader Impact Concerns:**

The broader impact concerns should address the intersection of the proposed methods and the problems of security such as, e.g.adversarial attacks (see Boloor et al (2019)).

Boloor et al (2019) Simple Physical Adversarial Examples against End-to-End Autonomous Driving Models, Arxiv

**Claims And Evidence:**

Yes

**Requested Changes:**

- the motivation statement improvement (see Weakness 1)
- addressing Weaknesses 2-4 on the references, experiments and clarity

The authors also need to address the following questions:
- The Section 5.1 mentions: “The hyper-parameters in A-adaptive LU (6) are chosen via sensitivity analysis, where λ = 0.01, τ = 0.1 for deblurring, and λ = 0.0001, τ = 0.1 for defogging.” Would the authors clarify whether such sensitivity analysis needs to be conducted for every new task?
- Is there any suggested procedure for designing the initial approximation operator A0, or is it limited to the assumption that it is some prior knowledge beyond the scope of the paper?
- The comparison is centred around the baseline of the Robust LU which may not be enough to conclude about superior performance; however, it would be interesting to see how the proposed model compares to the state-of-the-art deblurring such as  Zhang et al (2022), or the cited works such as Krainovic (2023)?
- It would be interesting to see how the choice of A0 impacts the performance, perhaps by varying the operator between the perfect match (A) and random parametrisation.
- In Table 1, is it possible to provide confidence intervals for the experimental results?
- Section 5.4 “This result aligns with the finding in Gilton et al. (2021a).” Is it possible to expand upon what was the finding in Gilton et al (2021a)?

**Strengths And Weaknesses:**

Strengths:
- The paper tackles important scenario of inverse problem beyond linear convolution model, which is an important open question
- The paper is well-written (with some aspects of clarity to be addressed)

Weaknesses:

(1) Motivation: the central question of the paper needs to be clarified. The paper’s title and abstract are centred around inverse problems in general; however, in the narrative one can see that it only focuses on image data (see, e.g., Figure 1), furthermore, it is not clear whether the authors want to centre the narrative around the problem of nonlinear inverse problems or robustness, therefore the authors should clearly state the benefits of the proposed scheme.

(2) References: Some of the recent approaches such as Zhang et al (2022) need to be cited; furthermore, it is important to show how this work fits into the wider range of inverse problems solving tasks (for images and beyond) such as Chung et al (2022).

(3) Experiments: While the approach permits non-linear models, it would be important to see what is the benefit of the proposed model against the linear counterpart, perhaps by direct comparison on the fog dataset.

(4) Clarity: it would be great to separate the related works section from the introduction. It would be also important to highlight which works have targeted the problem of nonlinear inverse problems.

Chung et al (2022) Improving Diffusion Models for Inverse Problems using Manifold Constraints, NeurIPS 2022

Zhang et al (2022) Pixel screening based intermediate correction for blind deblurring, CVPR 2022

---

> ### Author Response · Authors · 2024-03-18
> **Response to reviewer tfs5 (part 1)**
>
> We appreciate the thoughtful review by this respected reviewer. The reviewer's concerns are addressed point-by-point as follows.
>
> > Weakness (1): "the central question of the paper needs to be clarified. The paper’s title and abstract are centred around inverse problems in general; however, in the narrative one can see that it only focuses on image data (see, e.g., Figure 1), furthermore, it is not clear whether the authors want to centre the narrative around the problem of nonlinear inverse problems or robustness, therefore the authors should clearly state the benefits of the proposed scheme":
>
> Yes, the algorithm can be applied to non-imaging data. In fact, we would like to highlight that in the seismic deconvolution experiment, seismic data is time-sequential data collected from different sensor locations, which is non-imaging but can be visualized in 2D.
> We have discussed the benefits of our proposed method more clearly in the revised manuscript. Please refer to the paragraph (on page 2) just above the list of our contributions in the Introduction section, and the discussions of how our proposed works related to previous linear and nonlinear IP solvers with model mismatches in Related Works.
>
> In this work, we aim to underscore the relationship between handling model mismatch and the advantages of our proposed method in solving both linear and nonlinear inverse problems. The paper specifically addresses the challenge of model mismatch encountered when solving inverse problems with model-based architectures. Instead of learning to be robust to errors in the forward model implicitly from the learned regularizer in a model-based architecture, we propose a general framework that can actively handle the model mismatch in both linear and nonlinear IPs. The proposed framework holds the following benefits as compared to the existing active forward model mismatch algorithms:
> * more stable in training due to fewer hyperparameters introduced by the proposed general framework,
> * requires no additional data because no pretraining is required to learn the forward model,
> * enables simultaneous fitting of the forward model and reconstruction in a single pass.
>
> Please refer to the *Introduction* and *Related Works* sections for more detailed discussions.
>
> > Weakness (2): "Some of the recent approaches such as Zhang et al (2022) [1] need to be cited"
>
> We thank the reviewer for bringing this up. We have added the suggested works to the revised paper. Approaches in solving blind deblurring problems such as [1,2]  formulate the problem and regularization functions, particularly for blind deblurring tasks, which can be hard to generalize to nonlinear problems. We propose a general framework that can solve forward model mismatch for arbitrary inverse problems. In addition, a key distinction between [1] and our work is that our work allows a learned regularizer which is more powerful, rather than a pre-determined regularization function, which falls outside of the model-based inverse problem solvers.
>
> > Weakness (3): "While the approach permits non-linear models, it would be important to see what is the benefit of the proposed model against the linear counterpart, perhaps by direct comparison on the fog dataset."
>
> As suggested by the reviewer, we extend the experiments to landscape defogging using the linear counterpart of our proposed method (updating forward model mismatch using a matrix), and maintaining the proximal network architecture and the number of iterations. The linear estimated adaptive LU shows slight improvement over the robust LU, attributed to updates in the forward model, but is not as good as the performance of the proposed $\mathcal{A}$-adaptive LU. Please refer to the table given below for numerical comparisons. The result is expected because using a linear function to estimate a nonlinear forward model is suboptimal. A related discussion can be found in the last paragraph of the *Section 2.2 Methods for Addressing Model Mismatch* of the revised paper.
>
> | | Criteria | Defogging|
> | -------- | ------- | ------- |
> | Robust LU| PSNR | 29.645 $\pm$ 1.939 |
> | | SSIM | 0.9563 $\pm$ 0.0164   |
> | $\mathcal{A}$-adaptive LU (proposed method) | PSNR | **31.112** $\pm$ 1.854 |
> | | SSIM | **0.9661** $\pm$ 0.0133 |
> | Adaptive LU with linear estimation| PSNR | 29.732 $\pm$ 1.757 |
> | | SSIM | 0.9594 $\pm$ 0.0178 |
>
> Table: average and standard deviation of testing PSNR and SSIM for direct inverse mapping, robust LU, A-adaptive LU, and the linear estimate version of adaptive LU. The best performances for each task are in bold.

---

> ### Author Response · Authors · 2024-03-18
> **Response to reviewer tfs5 (part 2)**
>
> > Weakness (4): "it would be great to separate the related works section from the introduction. It would be also important to highlight which works have targeted the problem of nonlinear inverse problems."
>
> We have added the related works section to the original paper, including the discussions on model mismatch on both linear and nonlinear IPs. We would like to thank the reviewer for carefully reading the paper and highlighting the crucial structural changes that improved the paper significantly.
>
>
> > Would the authors clarify whether such sensitivity analysis needs to be conducted for every new task.
>
> Indeed, the hyperparameters for each task are determined based on the effectiveness of the initial forward model $\mathcal{A}$. In addition, in linear IPs, the eigenstructure of $\mathbf{A}$ also determines the residual approximated by $f_\theta = (\mathbf{A} - \mathbf{A}_0)x$, thus determining the choice of $\tau$.
>
> > Is there any suggested procedure for designing the initial approximation operator A0, or is it limited to the assumption that it is some prior knowledge beyond the scope of the paper?
>
> In this paper, we assume some prior knowledge of $\mathcal{A}_0$ is known, it is determined by simplified physics rule, approximated sensor locations, or approximation of parameters in the model. It is mentioned in Section 6 of the revised manuscript.
>
>
> > It would be interesting to see how the proposed model compares to the state-of-the-art deblurring such as [1].
>
> We have added this suggested work to the revised paper in the first paragraph of Section 2.2 Methods for Addressing Model Mismatch. The main distinction between [1] and our work is that [1] employs pre-determined (non-trainable) regularization functions, not a model-based architecture. While acknowledging other task-specific methods such as [1], our approach focuses on developing general model-based inverse problem solvers capable of handling forward model mismatch, rather than being tailored to specific tasks like blind deblurring. Our method is compared to the well-known and effective model-based architectures from an inverse problem's perspective, demonstrating performance gains when $\mathcal{A}$ is inaccurate. Additionally, [3] proposes a method of training a robust direct inverse mapping by adding random noise to the input $y$, which operates independently of knowledge about $\mathcal{A}$ and falls beyond the scope of addressing model-mismatch in model-based architectures. We have included the response in the revised manuscript in Section 2.2.
>
>
> > It would be interesting to see how the choice of $\mathbf{A}_0$ impacts the performance.
>
> Thanks for bringing up this interesting point. Among the three tasks demonstrated in this paper, image deblurring offers some flexibility in selecting an initial forward model. In seismic blind deconvolution, the deconvolution kernel is determined by estimating the wavelet from equipment. In landscape defogging, the forward model follows the relationship in equation (8) with unknown parameters $T$ and $L$. Thus, we extended the experiment in the Appendix (in supplementary material) to explore the effect of $\mathbf{A}_0$ in image blind deblurring reconstruction and demonstrated that our proposed methods are less sensitive to poor model initialization. Please refer to the Appendix of the revised manuscript for further discussions.
>
>
> > In Table 1, is it possible to provide confidence intervals for the experimental results?
>
> The standard deviation of testing PSNR and SSIM for each entry is added to Table 1 of the revised manuscript. We further found that the proposed $\mathcal{A}$-adaptive LU tends to have a smaller standard deviation than $\mathcal{A}$-adaptive DEQ because the latter does not have a fixed number of iterations. The changes are highlighted in Section 6.2 Reconstruction Results of the revised manuscript.
>
>
> > Is it possible to expand upon what was the finding in [4].
>
> This paper shows the convergence in reconstruction using DEQ when the forward model is precisely known. We observe a similar trajectory and highlight the convergence of our methods when updating the forward model iteratively. The changes are also added to Section 6.4.
>
> [1] Zhang et al (2022) Pixel screening based intermediate correction for blind deblurring, CVPR 2022
>
> [2] Liu et al (2021) Surface-aware blind image deblurring, IEEE Transactions on Pattern Analysis and Machine Intelligence
>
> [3] Krainovic et al (2023) Learning provably robust estimators for inverse problems via jittering. NeurIPS 2023
>
> [4] Gilton et al (2021) Deep equilibrium architectures for inverse problems in imaging. IEEE Transactions on Computational Imaging

---

> > ### Comment · Reviewer_tfs5 · 2024-03-26
> >
> > Many thanks for such a thorough response, I've carefully read it through as well as the discussion with the other reviewers. I think this addresses my concerns now, including on motivation, clarity and the paper structure.

---

### Decision · Action_Editor_L4Jw · 2024-04-23

**Recommendation:** Accept as is

**Comment:**

This paper considers model-based deep learning methods (such as loop unrolling (LU) and deep equilibrium model (DEQ)) for solving inverse problems. The authors solve a challenging problem where the forward model mismatch problem and introduce an untrained forward model residual block into the model-based methods, resulting in A-adaptive loop unrolling (LU) and A-adaptive deep equilibrium model (DEQ).  Convergence of A-adaptive DEQ has been proved under mild conditions. Experiments have been conducted to demonstrate a significant quality improvement in removing artifacts and preserving details for both linear and nonlinear inverse problems.

All the reviewers found that the claims in both theoretical analysis and practical experiments are well supported, illustrating the effectiveness and efficiency of the proposed architecture. They all recommend acceptance or leaning acceptance. I concur with the reviewers' recommendation to accept the paper.

**Audience:**

The result could be of interest to researchers working on the design of deep learning models, model-based deep learning methods, as well as their applications for inverse problems.

**Claims And Evidence:**

This paper considers model-based deep learning methods (such as loop unrolling (LU) and deep equilibrium model (DEQ)) for solving inverse problems. The authors solve a challenging problem where the forward model mismatch problem and introduce an untrained forward model residual block into the model-based methods, resulting in A-adaptive loop unrolling (LU) and A-adaptive deep equilibrium model (DEQ).  Convergence of A-adaptive DEQ has been proved under mild conditions. Experiments have been conducted to demonstrate a significant quality improvement in removing artifacts and preserving details for both linear and nonlinear inverse problems.